# Relationships of the Microbial Communities with Rumen Epithelium Development of Nellore Cattle Finished in Feedlot Differing in Phenotypic Residual Feed Intake

**DOI:** 10.3390/ani12070820

**Published:** 2022-03-24

**Authors:** Antonio M. Silvestre, Ana Carolina J. Pinto, Werner F. Schleifer, Lidiane S. Miranda, Leandro A. F. Silva, Daniel M. Casali, Katia L. R. Souza, Vanessa G. L. Gasparini, Gustavo D. Cruz, Garret Suen, Danilo D. Millen

**Affiliations:** 1School of Veterinary Medicine and Animal Science, São Paulo State University (UNESP), Botucatu 18618-000, Brazil; antoniosilvestrem@hotmail.com (A.M.S.); acjpinto@gmail.com (A.C.J.P.); 2College of Agricultural and Technological Sciences, São Paulo State University (UNESP), Dracena 17900-000, Brazil; werner.schleifer@outlook.com (W.F.S.); lidiane_mirand@outlook.com (L.S.M.); leandrosferreira02@gmail.com (L.A.F.S.); danielcasali2@gmail.com (D.M.C.); katia.li@outlook.com (K.L.R.S.); gasparini.v@hotmail.com (V.G.L.G.); 3Purina Animal Nutrition LLC, Arden Hills, MN 55126, USA; gdcruz@landolakes.com; 4Department of Bacteriology, University of Wisconsin-Madison, Madison, WI 53706, USA; gsuen@wisc.edu

**Keywords:** carcass, feed efficiency, Nellore, performance, ruminal epithelium, 16S ribosomal RNA

## Abstract

**Simple Summary:**

Several data have been published in the literature on the feed efficiency of cattle, measured through residual feed intake, in which the focus is to reduce production costs and improve cattle producer’s profit margins. Most of this variation in animal intake has been attributed to changes in animal metabolism, such as the rate of short-chain fatty acids absorption by the rumen epithelium, hepatic and mitochondrial metabolism, feeding behavior, and content of empty body fat. All these factors impact the anabolism/catabolism of substrates reaching the blood system. However, studies evaluating the composition of the rumen microbial community of animals differing in residual feed intake are scarce, and, therefore, this study aimed to compare the ruminal microbial communities, rumen morphometrics, feeding behavior, feedlot performance, and carcass characteristics of Nellore cattle, ranked by residual feed intake. Our results suggest that ruminal microbial community was not affected across different groups of residual feed intake, suggesting that differences in the efficiency of Nellore animals are associated with other factors.

**Abstract:**

The objective of this study was to examine the relationships among ruminal microbial community, rumen morphometrics, feeding behavior, feedlot performance, and carcass characteristics of Nellore cattle, classified by residual feed intake (RFI). Twenty-seven Nellore yearling bulls with an initial body weight (BW) of 423.84 ± 21.81 kg were fed in feedlot for 107 d in individual pens to determine the RFI phenotype. Bulls were categorized as high RFI (>0.5 SD above the mean, *n* = 8), medium RFI (±0.5 SD from the mean, *n* = 9), and low RFI (<0.5 SD below the mean, *n* = 10). At harvest, whole rumen content samples were collected from each bull to evaluate ruminal microbial community, including bacteria and protozoa. The carcass characteristics were determined by ultrasonography at the beginning and at the end of the experimental period, and behavior data were collected on d 88. As a result of ranking Nellore bulls by RFI, cattle from low-RFI group presented lesser daily dry matter intake (DMI), either in kilograms (*p* < 0.01) or as percentage of BW (*p* < 0.01) than high-RFI yearling bulls, resulting in improved gain:feed (G:F). However, variables, such as average daily gain (ADG), final BW, hot carcass weight (HCW) and other carcass characteristics did not differ (*p* > 0.05) across RFI groups. The eating rate of either dry matter (DM )(*p* = 0.04) or neutral detergent fiber (NDF) (*p* < 0.01) was slower in medium-RFI yearling bulls. For ruminal morphometrics an RFI effect was observed only on keratinized layer thickness, in which a thinner layer (*p* = 0.04) was observed in low-RFI Nellore yearling bulls. Likewise, Nellore yearling bulls classified by the RFI did not differ in terms of Shannon’s diversity (*p* = 0.57) and Chao richness (*p* = 0.98). Our results suggest that the differences in feed efficiency of Nellore bulls differing in phenotypic RFI should be attributed to metabolic variables other than ruminal microorganisms and epithelium, and deserves further investigation.

## 1. Introduction

Feed efficiency has been a variable of interest for many scientists in the last couple of decades, and it is related to the potential to improve animal performance and reduce production costs [1]. Likewise, the genetic selection of cattle presenting an improved feed efficiency is a reality worldwide [2]. It is well known that a variation in dry matter intake is present independent of body size and growth rate, and this variation was defined by Koch et al. [3] as residual feed intake (RFI), a measure of feed utilization efficiency by animals. Due to the improvement in feed efficiency in studies utilizing *Bos taurus* animals, a negative correlation was found between low-RFI animals and subcutaneous and intramuscular fat deposition [4,5]; however, for Nellore cattle, those responses were not confirmed [6,7,8].

Fat tissue deposition may negatively impact feed efficiency, since body fat has greater energy expenditure for its synthesis [9]. In addition, there is a very large variation in RFI across genetic groups, and even within breed itself, since the biological processes [1,10] involving these variations are not completely understood. Furthermore, the association of RFI variation in beef cattle with empty body composition seems to explain only a small part of the total variation in RFI [11]. Typically, cattle presenting high intakes, also present poor feed efficiency [12] due to their high energy intake, which would lead to greater deposition of fat. Higher intakes, especially in conditions where high-concentrate diets are fed, may predispose animals to metabolic disturbances [13], which may also negatively impact the efficiency of the animals.

As a result, a study was carried out to assess morphometric and histological characteristics of the rumen [14], in an attempt to explain the improved feed efficiency of low-RFI animals and its association with the absorptive capacity of rumen epithelium for products from fermentation. This has also been associated with the development of absorptive epithelium, in which cattle showing improved feed efficiency enhanced the absorption and metabolism of short-chain fatty acids (SCFA) [15], since these meet approximately 70% of the energy requirement of beef cattle [16]. These absorption processes have been related to increased mitochondrial activity due to transcriptional effects of proteins associated with oxidation, as well as epithelial integrity [17].

However, even though the absorptive capacity of the rumen plays an important role in the ability to use the fermentation products to determine feed efficiency of cattle, another variable that must be considered is the microbial community present in the rumen, which are responsible for the production rate of SCFA in the fermentation process. Therefore, studies associating RFI with the microbial community of the rumen may help to clarify some differences in microbial composition [18,19]. The microbial community of animals seems to be peculiarly intrinsic to each animal [20], in addition to what can be changed over the life of the animal according to changes in its diet [21]; therefore, this may help explain differences in animal feed efficiency [22]. The objective of this study was to examine the differences in the ruminal microbial communities between Nellore cattle groups differing in RFI, as well as the RFI effects on performance, carcass traits, ruminal traits and feeding behavior.

## 2. Materials and Methods

### 2.1. Description of Animals, Feeding and Management

The study was conducted using twenty-seven 20-month-old Nellore yearling bulls (423.84 ± 21.81 kg) fed in individual pens (6.0 × 12.0 m^2^) for 107 days with water available via water trough.

Cattle went through a receiving program, in which they were all de-wormed and vaccinated (tetanus, bovine viral diarrhea virus, 7-way Clostridium sp.; Cattle master and Bovishield, Pfizer Animal Health, New York, NY, USA). For the adaptation program for the finishing diet (85% concentrate), yearling bulls were stepped up through three adaptation diets containing 70%, 75%, and 80% concentrate for 6, 3 and 5 d, respectively (Table 1). Diets were formulated according to The Large Ruminant Nutrition System [23]. The yearling bulls were fed ad libitum twice a day, at 0800 h (55% of total ration) and 1500 h (45% of total ration), targeting 1% to 5% orts. Therefore, cattle received the same diet in each phase and were transitioned at the same time through adaptation up to the finishing diet.

The measurement of daily dry matter intake (DMI), expressed in kilograms and as a percentage of BW (mean BW), were performed by weighing what was offered daily (DM basis) minus the feed refusals of the next day (DM basis). At the beginning and at the end of the experiment, cattle were withheld from feed for 16 h to obtain initial and final BW. In order to estimate net energy both for maintenance and gain, the methods described by Lofgreen and Garrett [24], NRC (National Research Council) [25], and Zinn and Shen [26] were employed, and the relationship between these calculated values and the predicted values [23] was established.

At the beginning, and at the end of the experiment, longissimus muscle area (LMA), marbling content, biceps femoris fat thickness (P8), and 12th rib fat thickness were measured via ultrasound, as proposed by Perkins et al. [27]. Images were taken and analyzed for a single trained technician using an ultrasound containing the following specifications: Aloka SSD-1100 Flexus RTU unit (Aloka Co. Ltd., Tokyo, Japan) with a 17.2-cm, 3.5-MHz probe. The animals were slaughtered in a commercial abattoir using a captive bolt device The HCW was obtained after KPH fat removal, and the dressing percentage was calculated by dividing HCW by final BW.

### 2.2. Determination of Treatment by Divergence of RFI Groups

The study was planned as a completely randomized design, in which animal was considered the experimental unit. The experimental treatments were determined at the end of the experiment considering the data observed for metabolic BW (BW^0.75^) and ADG of the animals to obtain the RFI, which was calculated by the residue of the regression equation of observed DMI as a function of average daily gain (ADG) and average met-abolic BW (BW^0.75^) [28]:DMI = β0 + β1 × BW^0.75^ + β2 × ADG + ε,(1)
where β0 is the y intercept, β1 is the partial regression coefficient of midtest BW^0.75^, β2 is the partial regression coefficient of ADG, and ε is the error term.

Average daily gain during the test period was calculated by the difference between the initial and final weight, and the average between these two was used to calculate the BW^0.75^ and the G:F (overall ADG/overall DMI).

Residual BW gain (RG) was calculated as proposed by Crowleys et al. [29], where it was determined as the residual of the observed ADG regression equation as a function of DMI and average BW^0.75^:ADG = β0 + β1 × BW^0.75^ + β2 × DMI + ε,(2)
where terms are the same as described above, with the exception that β2 is the partial regression coefficient of DMI, and ε is the error term.

The Kleiber ratio was obtained by dividing ADG by average BW^0.75^ [30].

### 2.3. DMI Variations

Daily DMI variation was determined by the difference in intake between consecutive days throughout the study [31]. Daily DMI variation was tabulated and then expressed as percentage of DMI (considering absolute values) according to the equation below, as well as in kilograms:DMI variation, % = [(DMIcurrent day (kg) − DMIprevious day (kg))/DMIprevious day (kg)] × 100(3)

### 2.4. Feeding Behavior and Particle Sorting

The collection of data on feeding behavior took place on day 88 of the study following a method adapted from Robles et al. [32]. Data were recorded every 5 min during a 24-h period for each animal considering time spent eating, ruminating, resting (expressed in minutes), and number of meals per day, where a meal was considered the noninterrupted time cattle stayed in the feed bunk eating the ration.

On the day of the feeding behavior, the collection of samples of diets and orts were collected for chemical analysis of DM and NDF [33] to determine: intake of DM, intake of NDF, eating rate of DM (ERDM) and NDF (ERNDF), and rumination rate of DM (RRDM) and NDF (RRNDF), which were expressed in minutes per kilogram of either DM or NDF.

The Penn State Particle Size Separator was used for determination of particle-size distribution [34]. Particle sorting was determined as follows: n intake/n predicted intake, in which *n* = particle fraction screens of 19 mm (long), 8 mm (medium), 1.18 mm (short), and a pan (fine). Selective consumption values equal to 1 indicate no sorting, those <1 indicate selective refusals (sorting against), and those >1 indicate preferential consumption (sorting for).

### 2.5. Ruminal and Cecum Morphometrics

At harvest, scores for rumenitis and cecum lesions were performed in washed epitheliums after cattle evisceration. Rumen and cecum epithelium were classified according to Bigham and McManus [35] using a scale from 0 (no lesions and abnormalities noted) to 10 (severe ulcerative lesions). All rumens and cecum were scored by two trained individuals, who were blinded to the treatments.

The 1 cm^2^ samples of the ruminal epithelium were collected, as described by Resende Júnior et al. [36] and placed into a 70% alcohol solution for macroscopic measurements, such as: number of papillae per square centimeter of rumen wall (NOP); mean papillae area (MPA) and rumen wall absorptive surface area (ASA), which was calculated as follows: 1 + (NOP × MPA) − (NOP × 0.002).

For microscopic evaluations, 1 cm^2^ samples from both rumen and cecum epithelium were collected as described by Odongo et al. [37]. Histological rumen measurements, such as papillae height, papillae width, papillae surface area, keratinized layer thickness, and the mitotic index, were determined in 10% of total papillae population, based on NOP, per animal using computer aided light microscope for image analysis. For the mitotic index, the number of cells exhibiting mitotic figures was determined using the same microscope just described, and the final data were expressed as a percentage of 2000 cells. Histological measurements of the cecum were adapted from measurements of the rumen as described by Devant et al. [38] and Pereira et al. [39]. An electron light microscope with a Leica Qwin Image Analyzer was used to evaluate: crypt depth, goblet cells, enterocytes and crypt depth:goblet cells.

### 2.6. Collection and Preparation of Rumen Samples

The samples used for counting protozoa, and to study the ruminal bacterial com-munity composition, were collected right after slaughter, by opening the rumen of the animals. The preparation, counting and differentiation of protozoa was performed according to the methodology described by Pinto et al. [40], in which samples were analyzed using a Neubauer Improved Bright-Line counting chamber (Hausser Scientific Partnership R, Horsham, PA, United States) with optical microscopy (Olympus CH-2 R, Japan; Dehority, 1993) [41]. Protozoa were differentiated by genus: Isotricha, Dasytricha, Entodinium, and Diplodinium. The samples intended for the sequencing of the microbial community were collected by a trained person, and properly prepared with gloves to avoid contamination. A 50-mL amount of the rumen contents was then collected, including solid and liquid phases, and stored in tubes free of DNA and RNA, which were stored at −80 °C.

After thawing, samples were processed to isolate DNA following the procedure detailed in Weimer et al. [20] and Pinto et al. [40]. The microbial DNA was amplified using the PCR technique [41], in which, were used universal primers amplifying the 4 variable regions of the bacterial 16S rRNA gene (F-GTGCCAGCMGCCGCGGTAA; R-GGACTACHVGGGTWTCTAAT), as described by Kozich et al. (2013) [3]. Sequences were demultiplexed according to their sample-specific indices on an Illumina MiSeq, and deposited into the National Center for Biotechnological Information’s Short Read Archive, and is available under BioProject Accession PRJNA641164. Before reading, the samples were cleaned, as described by Pinto et al. [41], for removal of non-bacterial DNA contamination, and the software Mothur (v. 1.41.1) was used for further processing [42].

The SILVA 16S rRNA gene reference alignment database (v132; Quast et al., 2013) [43] was used to screen for alignment to the correct region. Pre-clustering was performed (diffs = 2) to reduce error and chimeras were detected and removed using UCHIME [44]. The GreenGenes database [45], August 2013 release, was used to classify sequences with a bootstrap value cutoff of 80. Sequences classified to cyanobacteria, mitochondria, Eu-karya, or Archaea were removed. Single-tons were removed to streamline analysis.

### 2.7. Statistical Analysis

The variables evaluated in the experiment were analyzed using the MIXED procedure od SAS (SAS 2003; SAS Institute, Cary, NC, USA) and the Tukey’s test to compare means, which was considered significant at *p* < 0.05. Tests for normality and heterogeneity of variances were performed prior to data analysis. An initial measurement covariate was added to the model when appropriate (*p* < 0.05).

Bacterial sequences were grouped into operational taxonomic units (OTUs) at 97% sequence similarity. Good’s coverage [46] was calculated in Mothur for all samples, considering ≥0.95 as having sufficient sequencing depth. The OTU counts were normalized to 10,000 sequences per sample, and the normalized counts of OTUs by sample were used for further analysis. Alpha diversity was assessed using Chao’s [47] estimate of species richness and Shannon’s [48] diversity index. Differences in community diversity and richness were assessed by overall 2-way ANOVA in R v3.2.1 (R Core Team, 2011) [49]. Beta diversity was assessed by using non-metric multidimensional scaling to visualize differences between samples using the Bray–Curtis dissimilarity metric [50]. Changes in total community structure (relative abundance, Bray–Curtis metric) were assessed using permutational multivariate ANOVA (PERMANOVA) in R (vegan package; v 2.5-2) [51]. Pairwise comparisons between each group were quantified by PERMANOVA, and *p*-Values were FDR(false discovery rate)-corrected.

## 3. Results

The regression equation developed for DMI in this study was fitted without intercept, given that it was not significant (*p* = 0.23). The initial 12th rib fat and P8 fat thickness were also tested in the model (*p* = 0.61 and *p* = 0.39 respectively). The final fitted equation was:DMI (kg/d) = 0.0544 × BW^0.75^ + 2.9659 × ADG; r^2^ = 0.996.(4)

Based on the residues obtained with the observed DMI data versus predicted DMI by Equation (4), animals were divided into: high-RFI group (8 animals), medium-RFI group (9 animals) and low-RFI group (10 animals).

### 3.1. Feedlot Performance and Carcass Characteristics

The residuals between the values predicted and observed by the regression for each animal are shown in Figure 1. The average of the residues for the high, medium and low RFI treatments were: 0.74, −0.04 and −0.58, respectively. The difference in DMI between high- and low-RFI yearling bulls observed in this study was 1.280 kg daily on average. Low-RFI animals (more efficient) consumed −0.58 kg/d, and high-RFI animals (less efficient) consumed + 0.74 kg/d for similar ADG (Table 2).

In order to justify the classification by the intake of these animals, those in the high-RFI group had higher DMI expressed in kilograms (*p* < 0.01) and as a percentage of BW (*p* = 0.02); however, DMI was similar when cattle from medium- and low-RFI groups were compared (*p* > 0.05). The low-RFI animals improved G:F when compared to high- and medium-RFI groups. Likewise, RG was lower for high-RFI group (*p* = 0.03) than for low-RFI animals; however, the Kleiber ratio was not different across treatments (*p* = 0.45).

### 3.2. Feeding Behavior and Selective Consumption

A significant effect was found for time spent resting (*p* = 0.03), time spent ruminating (*p* < 0.01), DMI (*p* = 0.05), NDF intake (*p* < 0.01), ERDM (*p* = 0.04), RRDM (*p* = 0.03), and ERNDF (*p* = 0.04). Cattle from high- and low-RFI groups had similar time spent resting and time spent ruminating; however, medium-RFI animals spent longer periods resting than high-RFI cattle, and also spent more time ruminating than animals from other treatments. Additionally, Nellore yearling bulls presenting medium-RFI had lower NDF intake and took longer to consume a kilogram of DM (*p* = 0.04) and NDF (*p* < 0.01). The time spent to ruminate a kilogram of DM was shorter for medium-RFI cattle compared to animals from high-RFI group (*p* = 0.03), but not different from low-RFI animals (*p* > 0.05). Same trend was observed for daily DMI (*p* = 0.045). No significant RFI effect was observed for particle sorting (Table 3) when diet and orts samples were sieved on a 19 mm (*p* = 0.47; long), 8 mm (*p* = 0.11; medium), 1.18 mm (*p* = 0.79; short) screens, and pan (*p* = 0.59; fine).

### 3.3. Rumenitis and Rumen and Cecum Morphometrics

The macro- and microscopic rumen morphometrics results across different RFI groups of animals did not differ in relation to their development (Table 4); however, there was an effect of the RFI grouping on keratinized layer thickness, in which animals from low-RFI group presented thinner layer (*p* = 0.04). As in the rumen, the cecum data showed no differences in development or lesions across the RFI groups.

### 3.4. Ruminal Protozoa Counting

Results of protozoa populations are shown in Table 5. No significant difference across treatments were observed for the genus *Diplodinium* (*p* = 0.60), *Isotricha* (*p* = 0.14) and *Dasytricha* (*p* = 0.68). However, cattle from the medium-RFI group presented a smaller population of *Entodinium* than high-RFI cattle (*p* = 0.01) without negatively impacting total protozoa populations (*p* = 0.27). When expressed as percentage of total population, no RFI-grouping effect was observed across treatments for the protozoa populations evaluated in this study (*p* > 0.05).

### 3.5. Ruminal Bacterial Community Composition

In the ruminal samples, collected right after slaughter for 16S rRNA microbiota sequencing, were generated a total of 76,510 raw sequences, resulting in an average of 5.77 ± 47 SD sequences per sample that passed filter. The pooled samples contained an average of 1.62 unique OTUs, and a Good’s coverage of 0.98.

The results related to bacterial community composition are shown in Figure 2. No significant RFI-grouping effect was observed for Shannon’s diversity (*p* = 0.57; Figure 2A) and Chao’s richness (*p* = 0.98; Figure 2B). Figure 2C shows the seven main phyla found in the ruminal content of animals differing in phenotypic RFI; however, no differences across treatments were detected within each phylum (*p* > 0.10). It is noteworthy to mention that at the time of slaughter, phyla Bacteroidetes and Firmicutes stand out. Finally, a Bray–Curtis dissimilarity analysis was conducted, as visualized using Non-metric multidimensional scaling (NMDS) (Figure 2D), and no differences across treatments were observed (*p* = 0.98).

## 4. Discussion

The main performance variables, such as final BW, ADG, HCW, dressing per-centage, as well as other carcass characteristics, were not affected by different RFI groups. Similar results involving Nellore bulls were reported by Nascimento et al. [52] and Fidelis et al. [53], which confirms that RFI in cattle is phenotypically independent of growth and body size, as defined by Koch et al. [17]. When the residual weight gain was calculated, cattle from low-RFI group exhibited an increase of 184 g when compared to high-RFI animals.

Data from this study show that there was no difference across all RFI groups regarding carcass traits measured by ultrasound, corroborating data reported by Nascimento et al. [52] and Fidelis et al. [53]. On the other hand, in a study by Pereira et al. [14], cattle from low-RFI group had thinner 12th rib- and P8-fat thickness by 0.86 mm and 0.88 mm, respectively, at harvest, as well as 680 g less visceral fat.

The body composition of animals has a great impact on energy requirements and daily gain composition [54]. In the literature, there are several studies that associated RFI variations in beef cattle with protein metabolism [1,5,9,55], which suggests that low-RFI animals have more efficient mechanisms related to protein metabolism, since these animals have leaner carcasses, lesser visceral, subcutaneous and intramuscular fat, and greater contribution of protein to weight gain. Although in our study we did not measure visceral fat, the assessment of carcass 12th rib fat and P8 fat thickness, as well as marbling, showed that low-RFI animals did not have any reduction of carcass fat deposition, suggesting that efficiency may be attributed to metabolic processes of each animal.

Energy metabolism in the process of absorption in the rumen [17] and intestines [56] has also been reported for the variation in energy expenditure by animals, in which, depending on the substrate and the absorption site, there are different energy expenditures for the animals [57]. The data from rumen morphometry, with the exception of keratinized layer thickness, did not reveal any significant difference between the RFI groups, which is in agreement with the data presented by Pereira et al. [14]. The thinner keratinized layer showed by low-RFI animals may be related to the lower DMI presented by these animals, since a smaller amount of substrate entered the ruminal environment for fermentation. In addition, a more intense visceral biochemical work was reported in animals with higher DMI [56]. In a survey carried out by Kong et al. [17] an increase in the expression of genes associated with glycolysis and oxidative phosphorylation in the epithelium of low-RFI animals was observed, which suggests a greater production of energy by these animals. The same authors also reported an increase in the expression of genes involved in cell and protein renewal in the modulation of intercellular adhesion force and cell migration, which, according to the authors, may be due to an increase in the absorption rate or to a greater surface area/volume ratio.

Although the visceral metabolism was not evaluated in this study, and the composition of the microbial community also did not differ between the RFI groups, this suggests that the differences in the efficiency of nutrient utilization in the different RFI groups may be related to the splenic and hepatic metabolism.

With respect to the rumen, the efficiency of the ruminal epithelium in removing the fermentation products can influence the ruminal acidification. In our data, animals from low RFI group consumed a 1.28 kg (DM basis) less than high-RFI cattle, which might explain the thinner keratin layer of the ruminal epithelium; however, further research is needed to identify potential differences on absorption rate of short-chain fatty acids in cattle differing in RFI.

Differences in feeding behavior were subtle in this study, and those are related to the smaller amount of NDF consumed by medium-RFI animals. As a result, medium-RFI cattle spent less time ruminating and took longer to consume a kilogram of either DM or NDF, which is in agreement with Llonch et al. [58]. In addition, *Entodinium* populations may have been negatively impacted by the lack of a proper amount of long particles in the rumen to maintain pH at optimal levels for growth of those microorganisms [13]. Results from this study related to feeding behavior are similar to the ones reported by Bingham et al. [59] and Pereira et al. [14]. Reasons by which medium-RFI animals sort against long and medium diet particles remain unknown and deserve further attention.

The quantification of the protozoa population in samples collected from the rumen at harvest showed no differences between the RFI groups in the total protozoan population, so that, even with a variation in DMI of 1.28 kg between extreme RFI groups, the animals probably managed to maintain the ruminal pH in adequate conditions to maintain these individuals, since mainly the genera *Diplodinium* e *Dasytricha* are sensitive to low ruminal pH [60,61]. In addition, rumen protozoa are important in the control of rumen pH, since they act in the predation of rumen bacteria and engulfing starch granules; where the genus *Entodinium* is the most dominant protozoan in the rumen of cattle fed high-concentrate diets [62], where they can rapidly degrade starch, resulting in faster growth rates [63]. In this study, the only difference between protozoan counts was found between the high and medium RFI groups, which is explained by the greater amount of starch ingested by high-RFI animals.

Furthermore, with respect to the ruminal bacterial communities, no differences were found in relation to Shannon’s diversity and Chao richness, which is in agreement with data from McCann et al. [64] who found no differences in Shannon’s diversity across different RFI groups of Brahman bulls in pasture. In addition, Myer et al. [65] fed cattle a high-concentrate diet, and also did not find differences in Shannon diversity and Chao richness. However, Welch et al. [21] evaluated the impact of RFI groups on the composition of the fecal microbial community and reported greater Shannon’s diversity in more efficient animals (6.99 vs. 7.81) fed high-concentrate diets, but this difference disappeared when animals were fed forage-based diets. Based on the fact just presented, it seems that the variation in feed efficiency across different RFI groups could not be attributed to the rumen microorganisms. Although our sample collection was performed after the animals had gone through a period of fasting, which may have influenced the microbes, this occurred for all animals. In addition, the transport and handling itself may influence intake, metabolism and shift the composition of the microbiota just prior to harvest. It is noteworthy to mention that cattle in this study were slaughtered on the same day they were shipped, which reduces the fasting time before slaughter.

Another limitation that we found is that the collection of samples did not cover microbes from the rumen wall, since microbes should be extracted from a washed rumen surface; however, the vast majority of the degradation process and consequent production of SCFA to increase animal gain occurs through the action of the microbiota adhered to feed particles, so that the collection of whole rumen contents from each animal was managed to cover and represent the entire rumen compartment.

## 5. Conclusions

The composition of the ruminal bacterial community and total protozoa in samples collected right after slaughter was not impacted when Nellore bulls finished in feedlot were ranked in different RFI groups. Likewise, the main morphometric characteristics of the rumen and cecum were not affected in terms of epithelial development and presence of lesions. Thus, differences in feed efficiency of Nellore bulls fed high-concentrate diets differing in phenotypic RFI should be attributed to metabolic variables other than ruminal microorganisms and epithelium, which deserves further investigation.

## Figures and Tables

**Figure 1 animals-12-00820-f001:**
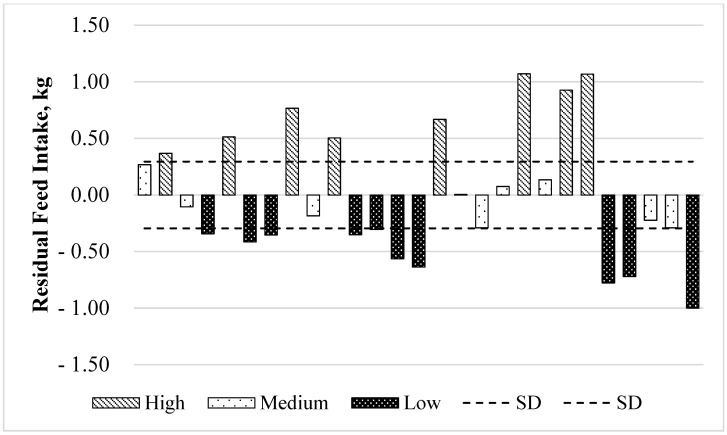
Residual feed intake for animals (*n* = 27) classified by BW^0.75^ and ADG by the following regression equation: DMI (kg/d) = 0.0544 × BW^0.75^ + 2.9659 × ADG. low RFI (<−0.579 kg/d; less than mean minus 0.5 SD), medium RFI (−0.039 to 0135 kg/d; ±0.5 SD of the mean) and high RFI (>0.743 kg/d; greater than mean plus 0.5 SD).

**Figure 2 animals-12-00820-f002:**
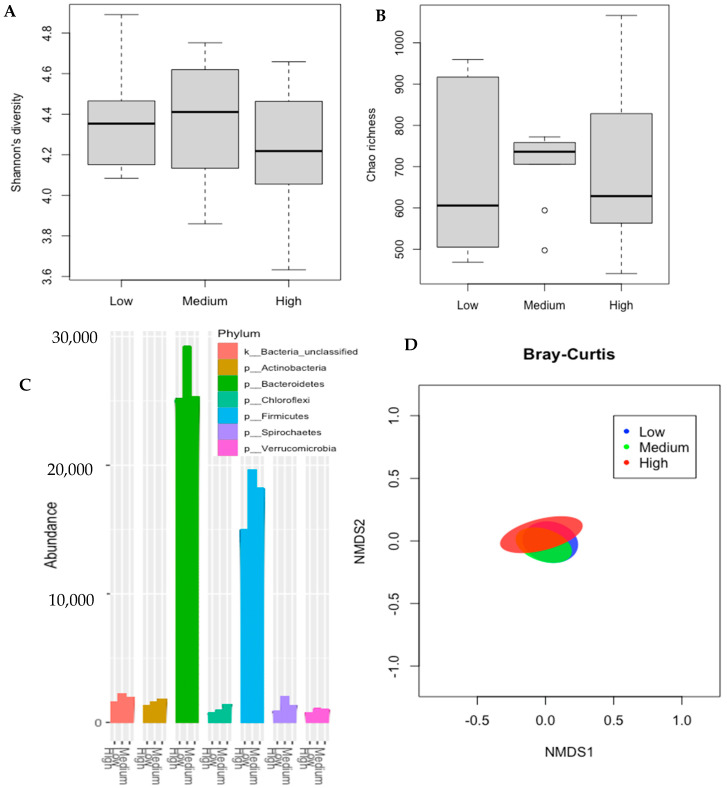
Shannon’s diversity index (**A**) and Chao’s richness estimator (**B**) for microbial communities in the rumen of Nellore cattle classified by RFI. Relative abundance of the top seven phylum of ruminal bacterial communities observed in different RFI groups Nellore cattle (**C**). Non-metric multidimensional scaling (NMDS) representation of the Bray–Curtis dissimilarity metric for ruminal content of Nellore cattle classified by RFI (**D**).

**Table 1 animals-12-00820-t001:** Feed ingredients and chemical composition of the experimental high-concentrate diets offered for all RFI-groups of Nellore yearling bulls (*n* = 27).

Diets	Adaptation 1	Adaptation 2	Adaptation 3	Finishing
**Level of concentrate, % of DM**	**70**	**75**	**80**	**85**
Sugarcane bagasse	18.0	16.0	14.0	12.0
*Cynodon dactylon* hay	12.0	9.0	6.0	3.0
Cracked corn grain	42.0	48.0	54.0	60.0
Citrus Pulp	8.0	9.5	11.0	12.5
Soybean meal	16.6	14.1	11.3	8.8
Supplement ^1^	3.4	3.4	3.7	3.7
Nutrient content, % of DM ^2^				
DM ^3^, % of OM ^4^	77.0	77.0	77.0	78.0
TDN ^5^	70.0	72.0	74.0	76.0
CP ^6^	16.1	15.5	15.1	14.5
NDF ^7^	34.8	31.7	28.6	25.4
peNDF ^8^	27.0	24.0	20.0	17.0
NEg (Mcal/kg of DM) ^9^	1.03	1.09	1.13	1.19
Ca	0.57	0.55	0.53	0.51
*p*	0.31	0.31	0.32	0.31

¹ Supplement contained for adaptation 1 and 2, and adaptation 3 and finishing, respectively, 26.47% and 44.44% of urea as a N source, 2.94% and 3.7% of mycotoxin adsorbent (Mycosorb; Alltech do Brasil Agroindustrial LTDA.; Araucaria, PR), 0.79% and 1.00% of monensin (Rumensin 200; Elanco saude animal Ltda, Baueri, SP, Brasil), as well as 69.79% and 50.85% of mineral supplement (DE Heus Industria e Comercio de Nutrição Animal LTDA; Rio Claro, SP) Ca: 24.0%; P: 2.0%; Mg: 2.5%; Na: 8.6%; S: 2.5%; Co: 24.0 ppm; Cu: 400.0 ppm; Fe: 30.0 ppm; Mn: 1000.0 ppm; Se: 8.0 ppm; Zn: 1800.0 ppm. ² Estimated by equations according to the Large Ruminant Nutrition System (Fox et al., 2004). ³ DM = dry matter. ^4^ OM = original matter. ^5^ TDN = total digestible nutrients. ^6^ CP = crude protein. ^7^ NDF = neutral detergent fiber. ^8^ peNDF = physically effective NDF. ^9^ NEg = net energy for gain.

**Table 2 animals-12-00820-t002:** Feedlot performance and carcass characteristics of Nellore bulls consuming high concentrate diets classified by residual feed intake (RFI).

Variable	RFI Group			SEM ^13^	*p*-Value
	High	Medium	Low		
*n*	8	9	10		
Feedlot Performance					
RFI ^1^, kg/d	0.74 ^a^	−0.04 ^b^	−0.58 ^c^	0.064	<0.01
Initial BW ^2^, kg	435	421	422	6.4	0.29
Final BW, kg	566	551	563	9.0	0.54
Daily DMI ^3^, kg	10.27 ^a^	9.09 ^b^	8.99 ^b^	0.270	<0.01
Daily DMI, % BW	2.06 ^a^	1.90 ^b^	1.83 ^b^	0.048	0.02
DMI Variation ^4^, %	6.86	6.10	6.64	0.420	0.34
DMI Variation, kg	0.60	0.54	0.55	0.025	0.27
ADG ^5^, kg	1.31	1.17	1.29	0.084	0.54
G:F ^6^, kg/kg	0.125 ^b^	0.128 ^b^	0.143 ^a^	0.0046	0.02
BW^0.75 7^, kg	105	104	105	0.7	0.54
RG ^8^, kg/d	−0.085 ^b^	−0.048 ^ab^	0.099 ^a^	0.0379	0.03
Kleiber ratio, g gain/kg	0.012	0.011	0.012	0.0007	0.45
Hot carcass weight, kg	307	297	308	4.3	0.22
Dressing percentage	54.51	53.89	54.40	0.371	0.44
NEm, Mcal ^9^/kg of DM	1.93	1.93	1.91	0.039	0.92
NEg, Mcal ^10^/kg of DM	1.29	1.28	1.26	0.034	0.92
NEm/NEm expected	1.05	1.05	1.03	0.021	0.87
NEg/NEg expected	1.06	1.06	1.04	0.028	0.86
Carcass characteristics					
Initial LMA ^11^, cm^2^	70.0	69.0	70.4	1.62	0.71
Final LMA, cm^2^	83.8	82.2	81.4	1.19	0.39
LMA daily gain, cm^2^	0.133	0.118	0.110	0.0114	0.39
Initial 12th rib fat, mm	3.1	3.1	3.0	0.15	0.87
Final 12th rib fat, mm	5.3	5.0	5.5	0.31	0.50
12th rib daily gain, mm	0.023	0.017	0.023	0.0028	0.32
Initial P8 ^12^ fat thickness, mm	4.3	4.4	4.0	0.25	0.50
Final P8 fat thickness, mm	7.3	6.9	6.9	0.39	0.87
P8 fat daily gain, mm	0.030	0.025	0.025	0.0032	0.47
Initial Marbling	2.1	2.4	2.1	0.21	0.59
Final Marbling	2.3	2.6	2.4	0.18	0.55

^1^ RFI = Residual feed intake. ^2^ BW = body weight. ^3^ DMI = dry matter intake. ^4^ DMI variation = difference between intake in consecutive days. ^5^ ADG = average daily gain. ^6^ G:F = gain to feed ratio. ^7^ BW^0.75^ = metabolic weight. ^8^ RG = residual weight gain. ^9^ Net energy for maintenance. ^10^ Net energy for gain. ^11^ LMA = longissimus muscle area. ^12^ P8 = fat thickness measured on biceps femoris muscle. ^13^ SEM = standard error of the mean. ^a,b,c^ For treatment effect, within a row, means without common superscript letter differ (*p* < 0.05).

**Table 3 animals-12-00820-t003:** Feeding behavior and particle sorting of Nellore yearling bulls consuming high-concentrate diets classified by residual feed intake (RFI).

Variable	RFI Group			SEM ^7^	*p*-Value
	High	Medium	Low		
*n*	8	9	10		
Feeding behavior					
Time spent resting, min/d	1001 ^b^	1092 ^a^	1032 ^ab^	21.8	0.03
Time spent ruminating, min/d	256 ^a^	155 ^b^	210 ^a^	17.5	<0.01
Time spent eating, min/d	182	189	197	8.6	0.43
Meal length, min	18	17	17	1.2	0.94
Meals per day, *n*	11	12	12	0.7	0.71
Water trough attendance, *n*/d	7	5	9	1.4	0.13
DMI per meal, kg	1.05	0.82	0.85	0.068	0.20
Daily DMI ^1^, kg	10.45 ^a^	9.13 ^b^	9.82 ^ab^	0.401	0.05
ER of DM ^2^, min/kg	18 ^b^	21 ^a^	20 ^b^	1.0	0.04
RR of DM ^3^, min/kg	25 ^a^	17 ^b^	22 ^ab^	1.9	0.03
NDF intake, kg	1.91 ^a^	1.51 ^c^	1.84 ^b^	0.087	<0.01
ER of NDF, ^4^ min/kg of NDF	97 ^b^	129 ^a^	109 ^b^	5.8	<0.01
RR of NDF, ^5^ min/kg of NDF	140	107	116	10.2	0.14
Particle sorting ^6^					
Long	0.80	0.69	0.65	0.061	0.47
Medium	0.94	0.88	0.95	0.028	0.11
Short	1.03	1.01	1.03	0.009	0.79
Fine	0.94	0.96	0.97	0.016	0.59

^1^ Daily DMI is the DMI of the day that behavior measurements were taken. ^2^ ER = eating rate of DM. ^3^ RR = rumination rate of DM. ^4^ ER = eating rate of NDF. ^5^ RR = rumination rate of NDF. ^6^ Selective consumption = *n* intake/*n* predicted intake, in which *n* = particle fraction screens of 19 mm (long), 8 mm (medium), 1.18 mm (short), and a pan (fine). Selective consumption values equal to 1 indicate no sorting, those <1 indicate selective refusals (sorting against), and those >1 indicate preferential consumption (sorting for). ^7^ SEM = standard error of the mean. ^a,b,c^ For treatment effect, within a row means without common superscript letter differ (*p* < 0.05).

**Table 4 animals-12-00820-t004:** Rumenitis and rumen morphometrics of rumen and cecum of Nellore yearling bulls consuming high-concentrate diets classified by residual feed intake (RFI).

Variable	RFI Group			SEM ^2^	*p*-Value
	High	Medium	Low		
*n*	8	9	10		
Macroscopic variables					
Rumenitis score	0.85	0.69	0.79	0.161	0.80
Number of papillae, no.	74.1	57.8	59.4	6.26	0.36
ASA ^1^, cm^2^/cm^2^ of rumen wall	41.1	32.2	33.9	3.65	0.25
Mean papillae area, cm^2^	0.56	0.55	0.56	0.036	0.97
Papillae area, % of ASA	97.5	96.8	97.2	0.32	0.42
Microscopic variables					
Papillae height, mm	1.56	1.52	1.60	0.078	0.72
Papillae Width, mm	0.17	0.18	0.17	0.005	0.80
Papillae surface area, mm^2^	0.25	0.23	0.25	0.015	0.54
Keratinized layer thickness, μm	32.8 ^a^	30.3 ^a^	28.2 ^b^	1.09	0.04
Mitotic index, %	6.36	6.26	6.13	0.574	0.96
Mitotic index, *n*	127	125	123	11.5	0.96
Cecum measurements					
Cecum score	2.60	1.57	3.24	0.730	0.25
Crypt depth, μm	109	96	105	5.7	0.30
Goblet cells, *n*	2.0	2.0	2.0	0.2	0.97
Enterocytes, *n*	20	19	22	0.7	0.12
Crypt depth:Goblet cells	61	52	59	6.3	0.62

^1^ ASA = absorptive surface area. ^2^ SEM = standard error of the mean. ^a,b^ For treatment effect, within a row means without common superscript letter differ (*p* < 0.05).

**Table 5 animals-12-00820-t005:** Differential counts of ciliated protozoa (10^3^/mL) of Nellore yearling bulls consuming high-concentrate diets classified by residual feed intake (RFI).

Variable	RFI Group			SEM ^1^	*p*-Value
	High	Medium	Low		
*n*	8	9	10		
Entodinium, 10^3^/mL	138 ^a^	120 ^b^	129 ^ab^	3.6	<0.01
Diplodinium, 10^3^/mL	80	76	75	4.6	0.60
Isotricha, 10^3^/mL	17	14	12	1.5	0.14
Dasytricha, 10^3^/mL	47	51	48	3.4	0.68
Total protozoa,10^3^/mL	281	262	263	8.4	0.27
Entodinium, %	49	47	49	1.2	0.42
Diplodinium, %	29	29	28	1.1	0.91
Isotricha, %	5.9	5.4	4.7	0.61	0.38
Dasytricha, %	17	20	18	1.0	0.13

^1^ SEM = standard error of the mean. ^a,b^ For treatment effect, within a row means without common superscript letter differ (*p* < 0.05).

## Data Availability

The data supporting reported results of Microbial Communities were deposited into the National Center for Biotechnological Information’s Short Read Archive, and is available under BioProject Accession PRJNA641164. Available upon request to the corresponding author danilo.millen@unesp.br.

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
