# Peer review of "Relationships of the Microbial Communities with Rumen Epithelium Development of Nellore Cattle Finished in Feedlot Differing in Phenotypic Residual Feed Intake"

_animals, 2022, doi:10.3390/ani12070820_

Round 1

Reviewer 1 Report

This paper attempts to relate rumen epithelium measurements with rumen microbiome and performance of beef cattle. This is a useful study, but there are limitations that will affect results and must be acknowledged and discussed fully. There are many other points that need to be considered.

BW was only measured once at beginning and end. BW varies from day to day and within a day. For accurate estimates of ADG, animals should be weighed regularly (daily or weekly) and a regression fitted. If animals are only weighed at beginning and end, they should be weighed at least twice each time. This must be discussed as a limitation.

Cattle were withheld from feed for 16 h to obtain initial and final BW. This will affect rumen microbiome, so findings might not apply to the fed animal. There are important implications. This must be discussed as a limitation.

Sampling limitations – rumen wall samples taken from cranial sac and ventral sac – why two places, any attempt to use same site each time? Rumen fluid – 50mL collected, which would not have included microbes attached to feed or rumen wall. There are important implications. This must be discussed as a limitation.

Title is opposite order from objective in abstract. Title implies there are associations, which there were not. Objective in abstract is not the same as in the main text (main text does not include rumen morphometrics). Revise title and objectives – make them consistent.

Too many decimal places in tables and text. This implies a high level of accuracy in measurements. For example, BW 423.84 kg suggests you can weigh animals to nearest 10g, Time spent resting 1000.83 min suggests you recorded to the nearest second. Only show three important numbers in each result (423 and 1000).

121 what value was used for “percentage of BW” – initial, final or mean BW?

151 Equation needs a subject (‘RIWG =’?)

155-160 The value and meaning of DMI fluctuation is not clear. Is this the same as DMI Variation in Table 2? Line 158 states that DMI fluctuation can be expressed “as a percentage of fluctuation”, and line 281 states that DMI fluctuation can be “expressed as percentage of variation”. Neither of these can be correct as the equation shows DMI fluctuation as a percentage of “DMI previous day”. Furthermore, results in Table 2 cannot be correct. Normal fluctuation results in animals eating more some days and less other days. That is why we take means over several days (e.g. week, month or trial) and assign a standard deviation or standard error. For animals in steady state, higher and lower intakes cancel each other, so average fluctuation is zero. For growing animals, we expect intake to increase over time, so average fluctuation is positive. In Table 2, average fluctuations are positive, but are biologically impossible. Overall mean fluctuation is +0.56 kg/d, so in 107 days, mean DMI would have increased by 60 kg/d.

Table 2 – what are NEm and NEg? These are measures of diet composition. If all animals were fed on the same diet, how can NEm and NEg possibly differ between groups? You can measure NE in vivo with calorimeters or by serial slaughter techniques, but neither were used in the current study. Explain what NEm and NEg are and correct the terminology, or delete.

247 Mothur must have a capital letter.

Table 3 – Units for ‘Time spent’ should be min/d? What is ‘Water trough per day’? No results for footnotes 6 and 7.

364-366 “no differences in ADG”, so you cannot pretend there was. Delete sentence.

366-371 – this is an unsound finding. There was no difference in ADG, and you classified animals according to RFI. Therefore, RFI-ADG (RIWG) is an artefact of your classification method. If retained, this statement needs a note of caution.

373 “Despite not significant, medium-RFI cattle sorted against long and medium diet particles” – you cannot write this if it was not significant.

413-416 Discussion about rumen pH, which was not measured. Why not?

433-436 not significant, so there was NO difference. Delete sentence.

Table 1 - All abbreviations must be explained in footnotes (for all tables).

Table 2 – What is ‘DMI Variation’? All abbreviations must be explained in footnotes.

Reviewer 2 Report

The following points need to be addressed thoroughly by the authors:

Material and Methods

  • The experimental design should be in a separate section with all the information required. Description of dietary treatments, number of replicates per treatment and number of animas in each replicate should be included.
  • What about the diversity of the animals. How this is taken into account, in the experiments? Is there any control group? Therefore the method used in the present study could introduce a significant bias in sequences output.
  • The method of sample preparation (e.g., equal weight of the rumen content from each animal being pooled) must be provided.
  • In addition, the human handlers can affect the gut microbiota composition; as this effect must be excluded before considering any significant difference. Therefore, section of samples collection/preparation should be included in the material and methods.

Results

  • Text material should highlight analysis or findings, summarizing only important details rather than reiterating the entire table or figure.

Discussion

The comparison of rumen bacterial community in RFI groups itself does not provide an informative knowledge to the international readers. The significance of the study must be more discussed.

Reviewer 3 Report

Please check the following sentences

Line 98-99   with water available via water trough (0.89 × 1.00 × 1.00 m).

Line 107 were fed ad li-bi-tum over

Define SEM on the tables

Line 373. If not statistical differences were found, please check this sentence

"Despite not significant, medium-RFI cattle sorted against long and medium diet particles, which certainly contributed to decrease NDF intake"

Statistical results may help to explain your results.

Aso, the last sentence should be revised or rewritten

In our results, although we did not find statistical difference, the abundance of bacteria in the phyla Bacteroidetes and Firmicutes was greater for medium-RFI animals, which may be related to the lower amount of Entodinium protozoa in the rumen of these animals, resulting in a reduction in the predation on bacteria.

You can not conclude with statistical support. Please check.

Round 2

Reviewer 1 Report

Authors have partially addressed the points raised. For major points, they have dismissed them or mentioned them briefly. They could have taken the opportunity to improve the paper with discussion of implications. It is good practice to discuss limitations of a study and their implications, instead of trying to hide them.

Problems with objectives and title have been addressed.

Authors have NOT addressed the problem of too many decimal places in tables and text. Their response is “We changed the decimal places according this reviewer’s suggestion”, which is incorrect. They did reduce decimals for time variables in Table 3, but all other numbers in that table need adjustment except NDF intake and particle sorting. All other tables include numbers with too many decimals, and there are many instances in the text (including the BW example I gave them). Only show three important numbers in each result.

DMI variation is now clear, but change ‘percentage of variation’ to ‘percentage of DMI’ in line 165. Change ‘DMI fluctuation’ to ‘DMI variation’ in line 167 (authors claimed they had done that).

Other minor comments have been addressed.

Reviewer 2 Report

The authors responded adequately to the raised points. The manuscript has been significantly improved and now permits publication in Animals.

Author Response

Thank you.

This manuscript is a resubmission of an earlier submission. The following is a list of the peer review reports and author responses from that submission.